# High-Protein Mulberry Leaves Improve Glucose and Lipid Metabolism via Activation of the PI3K/Akt/PPARα/CPT-1 Pathway

**DOI:** 10.3390/ijms25168726

**Published:** 2024-08-10

**Authors:** Ziyi Shan, Huilin Zhang, Changhao He, Yongcheng An, Yan Huang, Wanxin Fu, Menglu Wang, Yuhang Du, Jiamei Xie, Yang Yang, Baosheng Zhao

**Affiliations:** 1School of Life Sciences, Beijing University of Chinese Medicine, Beijing 100029, China; 2School of Chinese Materia Medica, Beijing University of Chinese Medicine, Beijing 100029, China; 3Beijing Research Institute of Chinese Medicine, Beijing University of Chinese Medicine, Beijing 100029, China

**Keywords:** high-protein mulberry leaves, traditional mulberry leaves, glucose metabolism, lipid metabolism, T2DM, obesity

## Abstract

High-Protein Mulberry is a novel strain of mulberry. High-Protein Mulberry leaves (HPM) were the subject of this study, which aimed to investigate its efficacy and underlying mechanisms in modulating glucose and lipid metabolism. A six-week intervention using *db*/*db* mice was carried out to assess the effects of HPM on serum lipid levels, liver function, and insulin (INS) levels. qRT-PCR and Western Blotting were employed to measure key RNA and protein expressions in the PI3K/Akt and PPARα/CPT-1 pathways. UHPLC-MS and the Kjeldahl method were utilized to analyze the component content and total protein. Additionally, network pharmacology was employed to predict regulatory mechanism differences between HPM and Traditional Mulberry leaves. The results of the study revealed significant improvements in fasting blood glucose, glucose tolerance, and insulin resistance in mice treated with HPM. HPM notably reduced serum levels of total cholesterol (TC), triglycerides (TG), low-density lipoprotein cholesterol (LDL-C), aspartate aminotransferase (AST), alanine aminotransferase (ALT), and INS, while increasing high-density lipoprotein cholesterol (HDL-C) levels. The treatment also effectively mitigated liver fatty lesions, inflammatory infiltration, and islet atrophy. HPM activation of the PI3K/Akt/PPARα/CPT-1 pathway suggested its pivotal role in the regulation of glucose and lipid metabolism. With its rich composition and pharmacodynamic material basis, HPM displayed a greater number of targets associated with glucose and lipid metabolism pathways, underscoring the need for further research into its potential therapeutic applications.

## 1. Introduction

Over the past several decades, countries worldwide have achieved significant advancements in modernization and urbanization. Consequently, detrimental habits such as staying up late, sedentary lifestyles, and unhealthy eating have become increasingly prevalent. This shift in lifestyle has been accompanied by a sharp rise in the incidence of type 2 diabetes mellitus (T2DM) and obesity. There is a complex interplay between obesity, T2DM, and cardiovascular diseases [1]. Research has indicated that obesity and T2DM significantly increase the risk of cardiovascular disease and stroke [2], with gender-specific differences in these indicators [3,4]. Insulin resistance, a phenomenon partly associated with obesity [5], is an early indicator of metabolic dysfunction. Most treatments for obesity and T2DM are interconnected. For instance, medications targeting obesity can also benefit diabetes management [6]. The urgent need for the development and promotion of drugs to combat T2DM and obesity is a global challenge.

Mulberry leaves, derived from *Morus alba* L., are extensively utilized in the clinical treatment of T2DM and obesity. Supported by the National Natural Science Foundation of China (NSFC), our team’s decade-long research has demonstrated the efficacy of mulberry leaves in regulating glucose and lipid metabolism. In this context, mulberry leaves will be referred to as Traditional Mulberry leaves for clarity.

High-Protein Mulberry, a strain newly cultivated by Chinese researchers, is now extensively planted [7,8]. Academic research on High-Protein Mulberry leaves (HPM) is scarce, primarily originating from agronomy, botany, and ecology. Recently, HPM has gained attention as animal feed, with research focusing on its applications in animal husbandry and its nutritional value. Studies have shown that incorporating HPM into the diet of mutton sheep significantly reduces blood glucose, lipid, and cholesterol levels [9]. Xiao et al. explored the production of HPM yogurt and found that it exhibited enhanced α-glucosidase inhibitory activity compared to plain yogurt [10], suggesting a potential transition from feed to food. Cao et al. [11] conducted a preliminary assessment of the active substance basis of HPM and Traditional Mulberry leaves using different extraction processes. Cao et al. [12] also investigated the protein extraction process and enzymatic degradation for hypoglycemic activity, identifying neutral protease enzyme degradation as the most effective against α-glucosidase inhibition. These studies provide valuable insights for the potential conversion of HPM from feed to pharmaceutical use. Previous research has only cursorily identified the regulatory effect of High-Protein Mulberry on glucose and lipid metabolism, but a comprehensive evaluation of its efficacy and mechanism is yet to be undertaken.

There is a valid scientific hypothesis that HPM can serve purposes beyond livestock feed, potentially offering greater value as a pharmaceutical intervention. HPM may possess a richer composition and content compared to Traditional Mulberry leaves, which could exert more significant medicinal effects on glucose and lipid metabolism. This experiment was designed based on this hypothesis.

## 2. Results

### 2.1. High-Protein Mulberry Leaves Downregulate Fasting Blood Glucose and Improve Obesity by Suppressing Intake

The food and water intake in the *db/db* mice was markedly higher compared to the Normal mice. Following administration of the test drug, there was a significant reduction in both food and water intake in the treated group compared to the Model mice (*p* < 0.05 or *p* < 0.01) (Figure 1a,b). Prior to drug administration, the *db/db* mice exhibited a substantially greater body weight than the Normal group. Post-intervention with HPM, the rate of weight gain in the *db*/*db* mice displayed a notable decrease (*p* < 0.05 or *p* < 0.01) (Figure 1c). Before treatment, the blood glucose levels in all *db*/*db* mice were above 11.1 mmol/L. Throughout the treatment period, the blood glucose levels in the Model group unfolded a gradual increase, confirming the efficacy of the model. The fasting blood glucose levels in the *db*/*db* mice were obviously higher than those in the Normal mice. However, the fasting blood glucose levels in mice treated with HPM demonstrated a pronounced decrease beginning from the fourth week of treatment (*p* < 0.05 or *p* < 0.01) (Figure 1d). Upon completion of the animal study, the liver index was calculated (Figure 1e). Concurrently, the body length of the mice was measured to determine the Lee’s index (Figure 1f). The liver index in the Model group was remarkably elevated compared to the Normal group (*p* < 0.01). In contrast, both the liver index and Lee’s index in all the treatment groups exhibited a marked reduction compared to the Model group (*p* < 0.01).

### 2.2. High-Protein Mulberry Leaves Ameliorate Fatty Degeneration of Liver and Protect Pancreatic Islets

Histological analysis of the liver and pancreas was conducted using H&E staining (Figure 2). The H&E staining results revealed that the liver cell structure in the Normal mice was intact and well-defined. The central vein was located at the center of the liver lobule, with hepatocytes and liver sinusoids arranged in a roughly radial pattern around it. The cytoplasm appeared loose, containing eosinophilic granules. The liver sinusoids exhibited no significant dilation or compression, and there was no apparent fibrosis or inflammatory cell infiltration. In stark contrast, the liver tissue of the Model mice displayed widespread severe watery and fatty degeneration of hepatocytes. This was characterized by cellular swelling, vacuolation, and a loose, lightly stained cytoplasm. The liver sinusoids were compressed, and the architecture of the liver cords and the boundaries of the liver cells were indistinct. Cytoplasmic lipid droplets were abundant, with occasional lymphocyte infiltration observed. Following six weeks of HPM intervention, only minor hepatocyte ballooning was evident in the treated mice. The cell bodies appeared slightly swollen with centralized nuclei, and a small amount of cytoplasmic vacuolation was observed. Local sinusoidal compression and some unclear liver cell boundaries were noted, but there was no significant inflammatory cell infiltration. The liver morphology in the HPM-treated group displayed marked improvement compared to the Model group.

In the pancreas, H&E staining also revealed the beneficial effects of HPM. The islets in the Model group exhibited a loss of regular shape and clear outline, with severe amyloidosis. In contrast, the pancreatic islets in the H-HPM group maintained a regular shape and clear outline, showing no obvious signs of atrophy, and amyloidosis was reduced. However, in the L-HPM group, the pancreatic islets were smaller in size, displayed significant atrophy, and had a reduced number of islet cells compared to the H-HPM group [13].

### 2.3. High-Protein Mulberry Leaves Actually Enhance Insulin Sensitivity and Modulate Glucose Metabolism

The results of the OGTT are presented in Figure 3a. Following the intragastric administration of glucose, the blood glucose levels in all groups rose rapidly within 15 min. The blood glucose levels in the Normal and Metformin groups began to decrease after 15 min, while the H-HPM, MLE, and L-HPM groups displayed a decline after 30 min. The blood glucose levels in the Model group decreased only after 90 min. Compared to the Normal group, the blood glucose levels in the Model group were markedly higher at each time point, as was the AUC, indicating a slower decrease in blood glucose (*p* < 0.01). In comparison with the Model group, the initial blood glucose levels in all the treatment groups were notably lower (*p* < 0.01). After 60 min, the blood glucose levels in these groups were decidedly lower than those in the Model group (*p* < 0.01). The AUC for H-HPM was apparently lower than that of the Model group, reflecting a more rapid decrease in blood glucose levels (*p* < 0.01). The ELISA results (Figure 3b,c) revealed that the serum insulin levels and the HOMA-IR index in the Model group were significantly higher compared to the Normal group (*p* < 0.01). In contrast, the serum insulin levels and HOMA-IR index in the H-HPM group were evidently lower than those in the Model group (*p* < 0.01). qRT-PCR analysis unfolded that the relative mRNA expression of *Pi3k* in the livers of mice in the Model group was noticeably reduced compared to the Normal group. However, this reduction was markedly reversed following treatment with H-HPM (*p* < 0.01) (Figure 3d). Western Blot analysis demonstrated that the ratio of p-Akt to Akt was distinctly decreased in the Model group. This alteration was notably reversed in the H-HPM group (*p* < 0.01) (Figure 3e). 

### 2.4. High-Protein Mulberry Leaves Effectively Regulate Lipid Metabolism and Improve Liver Function

Serum lipid profiles and liver function markers were assessed using biochemical assay kits, and the findings are depicted in Figure 4a. The Model-group mice exhibited manifestly elevated levels of TC, TG, and LDL-C compared to the Normal mice (*p* < 0.01), along with notably reduced levels of HDL-C (*p* < 0.01). Following six weeks of H-HPM treatment, the serum levels of TC, TG, and LDL-C in the mice transparently decreased (*p* < 0.01), while the HDL-C levels notably increased (*p* < 0.01). In comparison with the Normal group, the serum levels of ALT and AST in the Model group were predominantly higher (*p* < 0.01), indicating liver damage. However, this trend was reversed following drug intervention. The qRT-PCR results displayed that the relative expression of the *Cpt-1α* and *Pparα* genes in the livers of the Model-group mice was significantly lower compared to the Normal group (*p* < 0.01). In contrast, the relative expression of both *Pparα* (Figure 4b) and *Cpt-1α* (Figure 4c) genes in the livers of mice treated with H-HPM was significantly higher compared to the Model group (*p* < 0.01). Western Blot analysis (Figure 4d) corroborated these findings, revealing that the relative protein expression levels of CPT-1α and PPARα in the livers of the model group mice were notably lower compared to the Normal group (*p* < 0.01). However, the relative protein expression levels of CPT-1α and PPARα in the livers of mice treated with H-HPM were conspicuously higher compared to the Model group (*p* < 0.01).

### 2.5. High-Protein Mulberry Leaves Are Abundant in Their Composition and Exhibit a Significant Relative Content of the Primary Pharmacodynamic Material Basis

The UHPLC-MS analysis conducted in both positive and negative ion modes revealed distinct mass spectrometry profiles for HPM and Traditional Mulberry leaves (Figure 5a–d). These results indicated a compositional disparity between HPM and Traditional Mulberry leaves. A detailed examination of the spectral data allowed for the identification of the components present in each type of Mulberry leaf. The compilation of compounds from both ion modes provided a comprehensive list of the respective compositions found in HPM and Traditional Mulberry leaves (Appendix A). From this list, the known major pharmacodynamic constituents of Mulberry leaves, including 1-Deoxynojirimycin (1-DNJ), Chlorogenic acid, Cryptochlorogenic acid, and Rutin, were selected for further analysis. The proportions of these constituents were quantified by comparing them to the same volume of aqueous extracts from HPM and Traditional Mulberry leaves, respectively. The analysis revealed that the percentage of each of these four components in the aqueous extract of HPM was distinctly higher than that in Traditional Mulberry leaves (Figure 5e). Similarly, the protein content, as determined by the Kjeldahl method, followed the same trend. The protein content in the aqueous extract of HPM was found to be higher than that in Traditional Mulberry leaves (Figure 5f).

### 2.6. The Differential Components of High-Protein Mulberry Leaves Are Associated with a Larger Total Number of Genes Involved in T2DM and Obesity-Related Pathways

The differential components of High-Protein Mulberry leaves (HPM) and Traditional Mulberry leaves (MLE) were submitted to the Swiss Target Prediction database for target prediction, with a filter set at a probability greater than 0.1. This yielded 242 targets for HPM and 193 targets for MLE (Appendix A). Disease targets obtained from RStudio were combined with those retrieved from the GeneCards database, and any duplicates were eliminated, resulting in a total of 18,952 targets for T2DM and 16,255 targets for obesity. The construction of Venn diagrams from these disease targets versus the common and differential targets of HPM and MLE revealed 196 common elements among “T2DM”, “Obesity”, and “HPM”; 132 common elements among “T2DM”, “Obesity”, and “MLE”; and 215 common elements among “T2DM”, “Obesity”, and the common targets (Appendix A).

GO enrichment analysis was conducted on the aforementioned common elements among “T2DM”, “Obesity”, and the differential targets of HPM and MLE using the DAVID database, and the results are depicted in Figure 6a,b. KEGG pathway enrichment analysis for these common elements from Venn diagrams were individually illustrated in Figure 6c–e. The targets of the differential components of HPM and MLE that were enriched on each pathway were compiled, and the pathway-target interaction network was constructed using Cytoscape software 3.9.1 (Figure 6f,g).

Utilizing Centiscape 2.2, the PPI network for HPM initially comprised 180 targets and 855 edges. After screening, this network was refined to 32 core targets and 191 edges. Similarly, the PPI network for MLE initially had 127 targets and 598 edges, which was reduced to 28 core targets and 102 edges after screening. The respective core targets for HPM and MLE are presented separately in Figure 6h,i.

## 3. Discussion

Over the past few decades, T2DM has escalated into a global health crisis, posing a significant threat to human well-being [14]. The incidence and prevalence of T2DM and obesity among adolescents are also on the rise globally, with projections indicating a staggering 600 percent increase in young-onset T2DM by 2060 [15]. While there is a substantial body of research on antidiabetic medications, their side effects are a matter of concern [16,17]. Thus, there is an urgent need to explore safe and effective Chinese medicinal strategies for managing diabetes and obesity. Traditional Mulberry leaves have demonstrated efficacy in the treatment of metabolic disorders such as diabetes, dyslipidemia, obesity, atherosclerosis, and hypertension [18]. High-Protein Mulberry, a novel strain of Mulberry, has yet to undergo thorough pharmacological investigation. This study, therefore, initiated animal experiments using HPM as a pharmacological intervention, aiming to systematically analyze its efficacy and mechanisms in regulating glucose and lipid metabolism.

Diabetes is characterized by symptoms including polyuria, polydipsia, and polyphagia. The model group mice exhibited these symptoms, and treatment with HPM notably suppressed their appetite, as indicated by the line graph (Figure 1a,b), which is likely a contributing factor to the observed reduction in body weight. Lee’s index, often used as a measure of obesity in mice [19], showed a decrease in the HPM groups, reflecting an improvement in obesity.

Continuous apoptosis and necrosis of islet cells are one of the most pivotal causes of islet atrophy in diabetic mice. Islet amyloidosis is a complication of diabetes mellitus and is the only visible contributor to the decline in islet function. Some studies have displayed that islet amyloidosis is primarily caused by diabetes-related pathology [20]. Excessive precipitation and difficulties in clearing denatured proteins are important mechanisms in the progression of amyloidosis. The degree of amyloidosis is closely related to that of metabolic disorder [21]. It is revealed that amyloidosis in the H-HPM group has been improved by microscopic observation, which indicates that the progression of the diabetic condition is controlled to some extent after treatment with HPM. AUC of OGTT, serum insulin levels and HOMA-IR all notably increase in the model group, but evidently decrease after administration of HPM, which further demonstrates that HPM possess the effect of improving pancreatic beta cell function and alleviating insulin resistance. Insulin resistance is present in most individuals with T2DM with impaired glucose tolerance (IGT). In these cases, one way to prevent worsening of glucose tolerance is to increase insulin secretion, i.e., compensatory insulin secretion. In the measurement of insulin, the insulin level of the Normal group is considered as the Normal reference value, and the decrease in insulin level in the H-HPM group relative to the Model group indicates a reduction in the compensatory secretion of insulin, which in fact alleviates insulin resistance and enhances insulin sensitivity. The fact that HPM can regulate blood glucose is confirmed again. 

Disorders of glucose metabolism usually coexist with abnormalities of lipid metabolism in obesity-induced T2DM mice. The liver is an important target organ for the regulation of glucose and lipid metabolism, and hepatocellular damage is partly reflected in the levels of ALT and AST [22]. The four items of blood lipids are regarded as the barometers of the body’s lipid metabolism. The results of the biochemical kit and the histological analysis based on H&E staining indicate that the dosage of 8 g/kg of HPM is appropriate and does not cause any toxic or side effects on the liver, and at this dose HPM obviously improve the fatty degeneration of liver and the lipid accumulation. This provides reference for future laboratory research or clinical applications to some extent. 

The PI3K/Akt signaling pathway is a well-established signaling cascade that directly influences glucose metabolism by phosphorylating key metabolic enzymes [23,24]. The ratio of phosphorylated Akt to Akt serves as a marker for the activation status of the PI3K/Akt pathway. Activation of PPARα, a liver-predominant member of the peroxisome proliferator-activated receptors (PPARs), leads to reductions in triglyceride levels, ameliorates obesity, and inhibits liver fatty degeneration, thereby enhancing insulin sensitivity. PPARα plays a crucial role in the regulation of fatty acid oxidation, with fatty acid activation of PPARα promoting the oxidation of hepatic fatty acids to generate ketone bodies, which serve as an energy source for peripheral tissues [25]. PPARα further promotes fatty acid metabolism by modulating the expression of its downstream target CPT-1, thereby maintaining the lipid metabolism homeostasis within the body. The activation of the PI3K/Akt and PPARα/CPT-1 pathways in mice treated with HPM led to an obvious improvement in glucose and lipid metabolism disorders. These findings suggest that HPM promote the expression of genes and proteins involved in the breakdown of glucose and lipids, indicating a positive impact on the body’s metabolic health.

In the context of network pharmacology, a KEGG enrichment analysis was conducted on the common elements among T2DM, obesity, and the common targets of HPM and Traditional Mulberry leaves. The analysis revealed a high frequency of genes enriched in pathways such as the PI3K/Akt signaling pathway, lipid and atherosclerosis, cAMP signaling pathway, MAPK signaling pathway, and Hepatitis C, suggesting that the pharmacodynamic effects of HPM are likely to be associated with these pathways. Key genes in the PI3K/Akt and PPARα/CPT-1 pathways were selected for experimental validation, and the results confirmed that HPM regulate glucose and lipid metabolism through these pathways. 

Pioglitazone, akin to metformin, is also a medication employed in the management of T2DM [26]. Its primary mode of action is through the activation of PPARγ [27]. Upon activation of PPARγ by pioglitazone, a series of effects on glucose and lipid metabolism are initiated, which include enhancing insulin sensitivity, ameliorating insulin signaling, modulating lipid metabolism, and exerting anti-inflammatory actions. Consequently, in future research endeavors focusing on the mechanisms of glucose and lipid metabolism, particularly those centered on the activation of the PPAR pathways, the utilization of pioglitazone as a positive control is also commendable.

To delve more deeply into the mechanistic disparities between HPM and Traditional Mulberry leaves in the regulation of glucose and lipid metabolism, KEGG enrichment analysis was performed on the differential targets of HPM and Traditional Mulberry leaves in relation to T2DM and obesity. The analysis indicated that the pharmacodynamic differences might be linked to pathways such as the PI3K/Akt signaling pathway, insulin signaling pathway, nonalcoholic fatty liver disease, TNF signaling pathway, and FoxO signaling pathway. Specifically, HPM were found to be enriched for a greater number of genes associated with T2DM and obesity. KEGG enrichment analysis further revealed that targets such as MDM2, MAPK3, and PRKACB are associated with the PI3K/Akt and insulin signaling pathways, while TNF, MMP9, and MAPK3 are related to lipid metabolism. TNF is also implicated in inflammation-related pathways. The biological effects of insulin are integral to the PI3K/Akt signaling pathway, and lipid metabolism is intricately connected to lipids, atherosclerosis, and insulin resistance [28]. Abnormal lipid metabolism can lead to lipotoxicity, causing inflammatory reactions in tissues and organs [29]. Inflammation can inhibit insulin signaling, contributing to insulin resistance [30]. Thus, based on the predicted targets and KEGG pathway enrichment results, HPM likely regulate glucose and lipid metabolism by reducing insulin resistance, promoting lipid metabolism and exerting anti-inflammatory effects. The study’s limitation lies in the exploration of HPM’s mechanism being confined to classical glucose and lipid metabolism pathways, without investigating other predicted differential pathways. This will be a focus of future research endeavors.

## 4. Materials and Methods

### 4.1. Drug Preparation

The Traditional Mulberry leaves were supplied by Beijing Taiyang Shukang Pharmaceutical Group Co., Ltd. (Beijing, China) (lot number 2212057), while the High-Protein Mulberry leaves (HPM) were harvested from the Agricultural Ecological Park in Liming Village, Beijing. The HPM were meticulously rinsed and then dried in a controlled environment, free from humidity and light, at a constant temperature of 24 °C over a period of 14 d. Subsequently, both the dried HPM and Traditional Mulberry leaves were each weighed to an exact 1000 g. They were then soaked in ten times their weight of water for 30 min, followed by a 2 h decoction and filtration process. The remaining residue was resoaked with eight times its weight of water and subjected to another 2 h decoction and filtration. The resulting extracts were combined, concentrated using a water bath (model Kewei HH-S8A), and finally dried in a vacuum drying box (model Yiheng DZF-6092) to obtain a dry paste. The metformin was provided by Sino-American Shanghai Squibb Pharmaceuticals Ltd. (Shanghai, China) (lot number ACJ6699).

### 4.2. Animals and Groups

Thirty-five 7-week-old male BKS.Cg-Dock7*^m+^*^/*+*^ Lepr*^db^*/Nju (*db*/*db*) mice and seven 7-week-old male C57BLKS/J littermate wild-type (*m*/*m*) mice were obtained from Jiangsu Huachuang Xinnuo Pharmaceutical Technology Co., Ltd. (Taizhou, China)(license number SCXK (Jiangsu) 2020-0009). Metformin was selected as the positive control drug. Following a one-week adaptive feeding period, the *db*/*db* mice were randomly allocated into five groups of seven mice each based on blood glucose and body weight: the Model group, the Metformin group, the Traditional Mulberry leaves extract group (MLE), the high-dose High-Protein Mulberry leaves extract group (H-HPM), and the low-dose High-Protein Mulberry leaves extract group (L-HPM). Seven *m*/*m* mice were included as the Normal group. The Model was deemed successfully established if the *db*/*db* mice maintained stable blood glucose levels above 11.1 mmol/L before initiating drug administration. The clinical dosage range for mulberry leaves is 15–60 g [31], with 60 g being the maximum human dose, equating to approximately 8 g/kg for mice. The MLE and H-HPM were administered at a crude drug dose of 8 g/kg, the L-HPM at 4 g/kg, and the Metformin at 200 mg/kg. The drugs were administered intragastrically at a volume of 0.1 mL/10 g body weight. All the mice were provided with standard-maintenance feed and had ad libitum access to food and water throughout the study. The mice were monitored for 6 weeks under conditions of (23 ± 1) °C room temperature, 60~65% relative humidity, and a 12 h/12 h light-dark cycle. Body weight, blood glucose, food intake, and water intake were recorded weekly. The animal protocol was reviewed and approved by the Medical and Experimental Animal Ethics Committee of Beijing University of Chinese Medicine, with license number BUCM-2023031604-1070.

### 4.3. Oral Glucose Tolerance Test

The oral glucose tolerance test (OGTT) was conducted one week prior to the conclusion of the animal experiment. The mice were fasted for 12 h with water withheld and then were administered glucose orally at a dose of 2 g/kg. Blood samples were collected from the tail vein at 0, 15, 30, 60, and 120 min post-gavage to measure blood glucose levels. The area under the curve (AUC) of the OGTT was calculated for each group using the following formula, where BG represents blood glucose:AUC = 0.25 × (0.5 × BG0 min + BG15 min + 1.5 × BG30 min + 2 × BG60 min + 2 × BG90 min + BG120 min)

### 4.4. Sample Preparation

Upon completion of the animal experiment, blood was collected by ocular puncture, body length was measured to calculate Lee’s index, and tissue samples from the liver and pancreas were harvested. The blood samples were centrifuged at 4 °C for 15 min at 3000 rpm using an Eppendorf centrifuge (Hamburg, Germany), and the serum was then separated. All the samples were stored in a refrigerator at −80 °C. The Lee’s index was calculated using the following formula:Lee’s index = weight (g) ^ (1/3)/body length (cm)

### 4.5. Histological Analysis

Hematoxylin and Eosin (H&E) staining was conducted on the liver and pancreas tissues to examine histopathological alterations. Tissues fixed in 4% paraformaldehyde were processed for paraffin embedding, sectioned and subsequently dewaxed. Hematoxylin and eosin staining were then performed. Following staining, the slides were dehydrated, coverslipped, and analyzed under a light microscope (Olympus Corporation, Tokyo, Japan, Model CX23).

### 4.6. Biochemical Analysis

The levels of triglycerides (TG), total cholesterol (TC), high-density lipoprotein cholesterol (HDL-C), low-density lipoprotein cholesterol (LDL-C), alanine aminotransferase (ALT), and aspartate aminotransferase (AST) in the mice serum were measured using biochemical assay kits supplied by Jiancheng Bioengineering Institute, Nanjing, China. The insulin (INS) levels in the mice serum were quantified with an enzyme-linked immunosorbent assay (ELISA) kit (Kete, Yancheng, China), and the Homeostatic Model Assessment for Insulin Resistance (HOMA-IR) was calculated using the following formula:HOMA-IR = fasting blood glucose (mmol/L) × fasting insulin (mIU/L)/22.5

### 4.7. qRT-PCR Analysis

The key pathways in glucose and lipid metabolism—the PI3K/Akt pathway and the PPARα/CPT-1 pathway—were selected for analysis [32,33,34]. The expressions of critical genes and proteins within these pathways were chosen as measurement parameters. Total RNA was extracted and reverse transcribed according to the instructions provided with the kit (Beyotime, Shanghai, China). The qRT-PCR reaction mixture was prepared as follows: 2× ChamQ Universal SYBR qPCR Master Mix (10.0 μL), Forward primer (0.4 μL), reverse primer (0.4 μL), cDNA (1.0 μL), and ddH2O to a total volume of 20.0 μL. The mRNA content of the target genes was normalized to the internal reference gene *β-actin*, and the relative expression of target genes was determined using the 2^−ΔΔCT^ method. The primer sequences are presented in Table 1.

### 4.8. Western Blotting Analysis

The liver tissue was homogenized, and protein concentrations were determined using a BCA protein assay kit (NCMbiotech, Suzhou, China). Total protein (10 μg) was separated via SDS-PAGE and then transferred to PVDF membranes (Merck Millipore, Darmstadt, Germany). The membranes were incubated overnight at 4 °C with the following primary antibodies: Carnitine palmitoyl transferase 1 α (CPT-1α) (1:1000, ab234111, Abcam, Cambridge, UK), Peroxisome proliferator-activated receptor α (PPARα) (1:1000, ab126285, Abcam), Protein Kinase B (Akt) (1:10,000, ab179463, Abcam), Phospho-Protein Kinase B (p-Akt) (1:1000, ab192623, Abcam), and α-Tubulin (1:2000, ab176560, Abcam) as the loading control. After incubation with primary antibodies, the membranes were then probed with goat anti-rabbit IgG-HRP (1:5000, ab205718, Abcam) for 60 min at room temperature. Protein band signals were visualized using the ChemiDoc MP Imaging System (Bio-Rad, California, America), and quantitative analysis of the protein bands was performed using ImageJ software 1.54i (NIH). The expression levels of target proteins were normalized to α-Tubulin.

### 4.9. UHPLC-MS

A precise weight of 100 mg of the well-mixed sample was transferred into a 2 mL centrifuge tube, to which 1 mL of 70% methanol and 3 mm diameter steel beads were added. The sample was pulverized by agitation in a fully automated sample rapid grinder (Jingxin, Shanghai, China, model JXFSTPRP-48, 70 Hz) for 3 min. After allowing the mixture to cool, it was subjected to low-temperature sonication (40 KHZ) for 10 min, followed by centrifugation at 12,000× *g* rpm for 10 min at 4 °C. The supernatant was then collected and diluted 2–100 times. Ten μL of a 100 μg/mL internal standard solution was added, and the mixture was filtered through a 0.22 μm PTFE filter. 2-Amino-3-(2-chlorophenyl) propanoic acid (provided by Yuanye, Shanghai, China) was used as the internal standard in this experiment. The analysis was conducted using a Zorbax Eclipse C18 chromatographic column (1.8 μm × 2.1 mm × 100 mm, Agilent Technologies, California, America) under the following chromatographic conditions: column temperature of 30 °C; flow rate of 0.3 mL/min; mobile phase A consisting of 0.1% formic acid aqueous solution and mobile phase B being pure acetonitrile. The injection volume was 2 μL, and the autosampler was maintained at 4 °C. Positive and negative ions were detected separately using the following mass spectrometry conditions: heater temperature of 325 °C; sheath gas flow rate of 45 arb; auxiliary gas flow rate of 15 arb; purge gas flow rate of 1 arb; electrospray voltage of 3.5 KV; capillary temperature of 330 °C; S-Lens RF Level at 55%. The analysis employed a primary full scan (Full Scan, *m*/*z* 100~1500) and a data-dependent secondary mass spectrometry scan (dd-MS2, TopN = 5). The resolution was set at 120,000 for primary MS and 60,000 for secondary MS. The collision mode used was high-energy collisional dissociation (HCD).

### 4.10. Kjeldahl Method

The sample was weighed into a digestion tube, to which 0.4 g of copper sulfate, 6 g of potassium sulfate, and 20 mL of sulfuric acid were added. The mixture was then subjected to digestion in a furnace. Upon reaching a temperature of 420 °C, the liquid in the digestion tube turned blue-green, becoming clear and transparent. The digestion process was continued for an additional hour. After cooling, 20 mL of water was introduced into the solution. Subsequently, the solution was distilled using an automatic Kjeldahl nitrogen analyzer (Haineng, Jinan, China, K9840) for a duration of 7 min. In the receiving bottle, a mixture of indicators comprising 1 part methyl red ethanol solution (1 g/L) and 5 parts bromocresol green ethanol solution (1 g/L) was added, along with 10 mL of boric acid solution (20 g/L). The distillate was collected until it reached 200 mL, and then titrated with a standard hydrochloric acid solution (0.100 mol/L). The endpoint was indicated by a light grey-red color. A reagent blank was also performed simultaneously.

### 4.11. Network Pharmacology

The components identified by Ultra-High Performance Liquid Chromatography-Mass Spectrometry (UHPLC-MS) were utilized to generate Canonical Simplified Molecular Input Line Entry System (SMILES) codes via the PubChem database (https://pubchem.ncbi.nlm.nih.gov/ (accessed on 26 April 2024)). These SMILES codes were subsequently employed to predict potential targets using the Swiss Target Prediction platform (http://www.swisstargetprediction.ch/ (accessed on 26 April 2024)), with a filter applied to retain targets with a probability greater than 0.1. This process yielded the pharmacological targets associated with HPM and Traditional Mulberry leaves, respectively. The components discerned by UHPLC-MS, which are present in HPM but absent in Traditional Mulberry leaves, and those found in Traditional Mulberry leaves but not in HPM, are collectively designated as differential components in this context. Conversely, the components shared by both HPM and Traditional Mulberry leaves are classified as common components. The targets corresponding to these differential and common components are henceforth referred to as differential targets and common targets, respectively. The differential components and their respective targets for HPM and Traditional Mulberry leaves were compiled separately. For the purpose of this study, datasets related to “type 2 diabetes” and “obesity” were retrieved from the Gene Expression Omnibus (GEO) database, with the search criteria refined to include only “Homo sapiens” data and “series”. Following a comprehensive review of the literature, GSE83452 [35,36] was selected as the representative dataset for “obesity”, while GSE20966 [37,38] was chosen for “type 2 diabetes”. The gene symbol names from these datasets were converted using RStudio. Additional targets for “type 2 diabetes” and “obesity” were obtained from the GeneCards database (https://www.genecards.org/ (accessed on 28 April 2024)), with a filter for relevance scores exceeding 0.5. The compiled targets, along with those converted through RStudio, were merged to form a comprehensive set of disease-related targets. Venn diagrams were generated to visualize the overlap between the disease targets and the common and differential targets of HPM and Traditional Mulberry leaves. The intersecting targets from these diagrams were then used to construct protein–protein interaction networks (PPI) using the String database (https://cn.string-db.org/ (accessed on 30 April 2024)), with the species set to “Homo sapiens” and the results exported in TSV format. Concurrently, the intersecting targets were submitted to the DAVID database (https://david.ncifcrf.gov/ (accessed on 30 April 2024)) for KEGG pathway and Gene Ontology (GO) enrichment analysis. The PPI networks were imported into Cytoscape 3.9.1 for the construction of pathway-target interaction networks. To identify core targets within the differential components of HPM and Traditional Mulberry leaves, the Centiscape 2.2 plugin was employed, utilizing the betweenness, closeness, and degree as filtering parameters. For HPM, the core targets were identified with a betweenness value of 332.2555556, a closeness value of 0.00201353, and a degree value of 9.5. Similarly, for Traditional Mulberry leaves, the core targets were determined with a betweenness value of 191.3385827, a closeness value of 0.003227088, and a degree value of 9.417322835.

### 4.12. Statistical Analysis

All the experiments were conducted with a minimum of three replicates, and the data are presented as the mean ± standard deviation (SD). Statistical significance was evaluated using one-way analysis of variance (ANOVA) with GraphPad Prism 9 software. *p* < 0.05 was deemed statistically significant.

## 5. Conclusions

High-Protein Mulberry leaves, characterized by their rich composition and significant relative content of pharmacodynamic material, demonstrate efficacy in enhancing glucose and lipid metabolism. These leaves exhibit a higher number of targets associated with pathways critical for glucose and lipid regulation, making them a subject of significant research interest. High-Protein Mulberry leaves are regarded as a promising novel therapeutic agent, warranting further development and investigation to harness their full potential in the treatment of metabolic disorders.

## Figures and Tables

**Figure 1 ijms-25-08726-f001:**
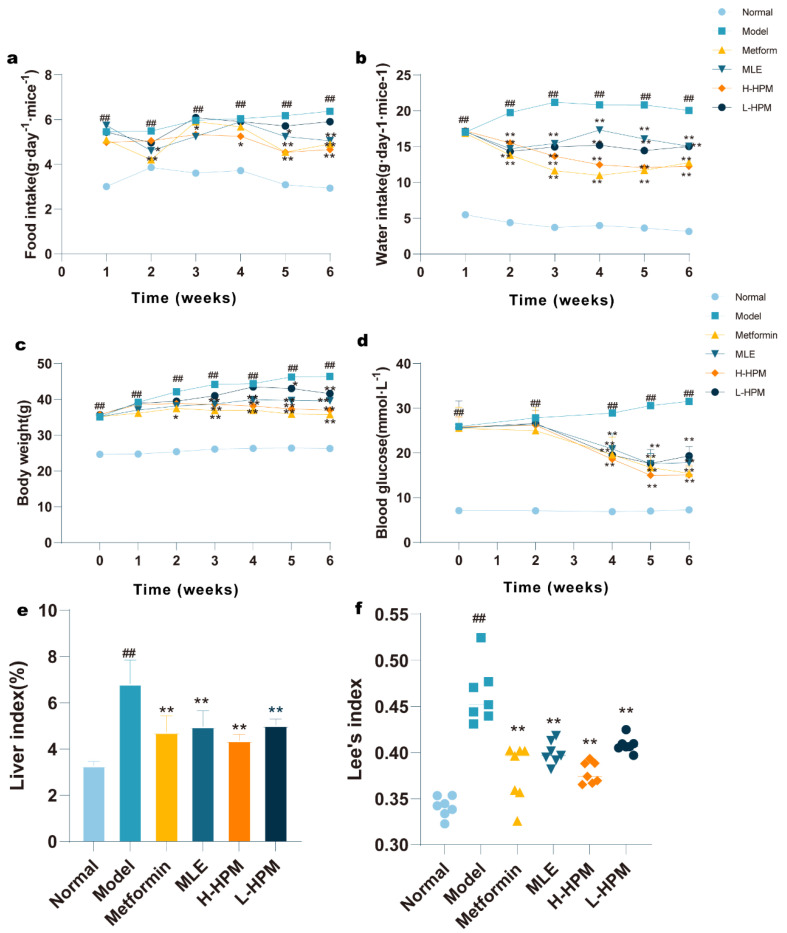
High-protein Mulberry leaves downregulate fasting blood glucose and improve obesity by suppressing intake. (**a**–**f**) Food intake, Water intake, Body weight, Blood glucose, Liver index, and Lee’s index of mice in each group. * *p* < 0.05, ** *p* < 0.01 vs. Model group; ^##^ *p* < 0.01 vs. Normal group. (n = 7).

**Figure 2 ijms-25-08726-f002:**
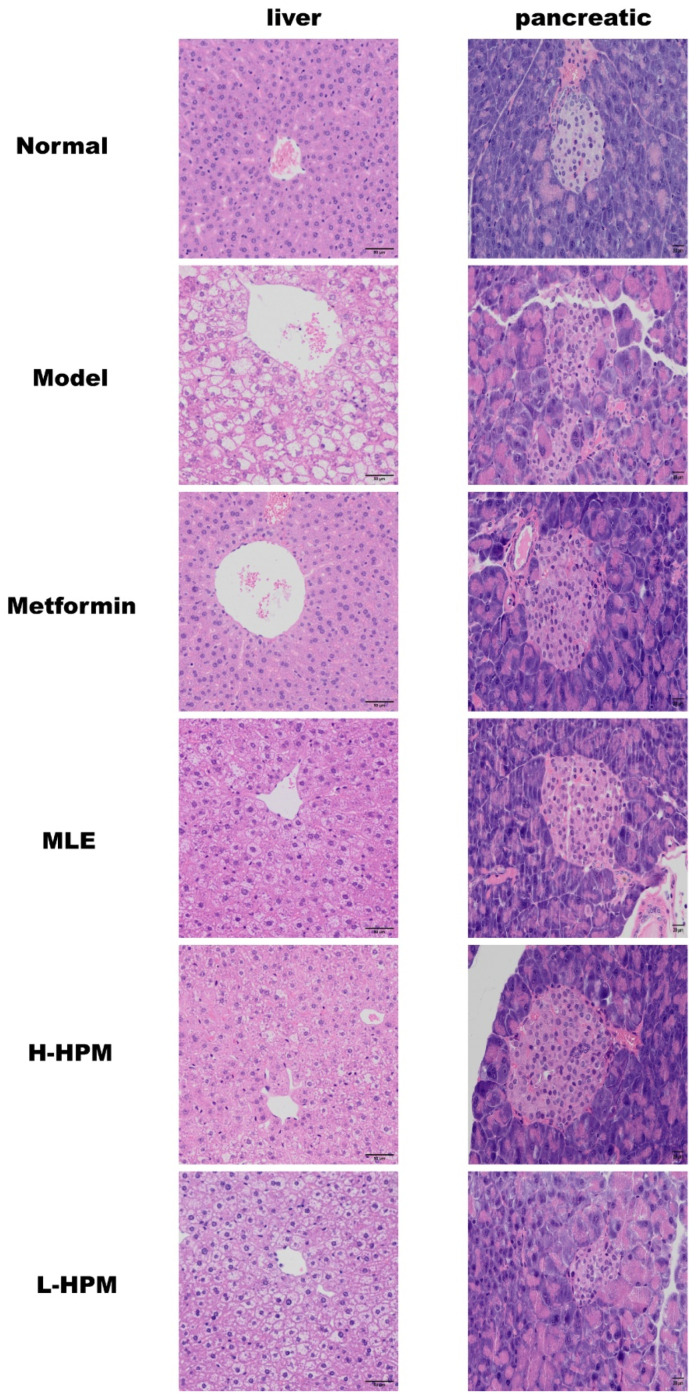
High-Protein Mulberry leaves ameliorate fatty degeneration of liver and protect pancreatic islets. The column on the left is the H&E images of liver, Scale bar = 50 μm; the column on the right is the H&E images of pancreatic, Scale bar = 20 μm.

**Figure 3 ijms-25-08726-f003:**
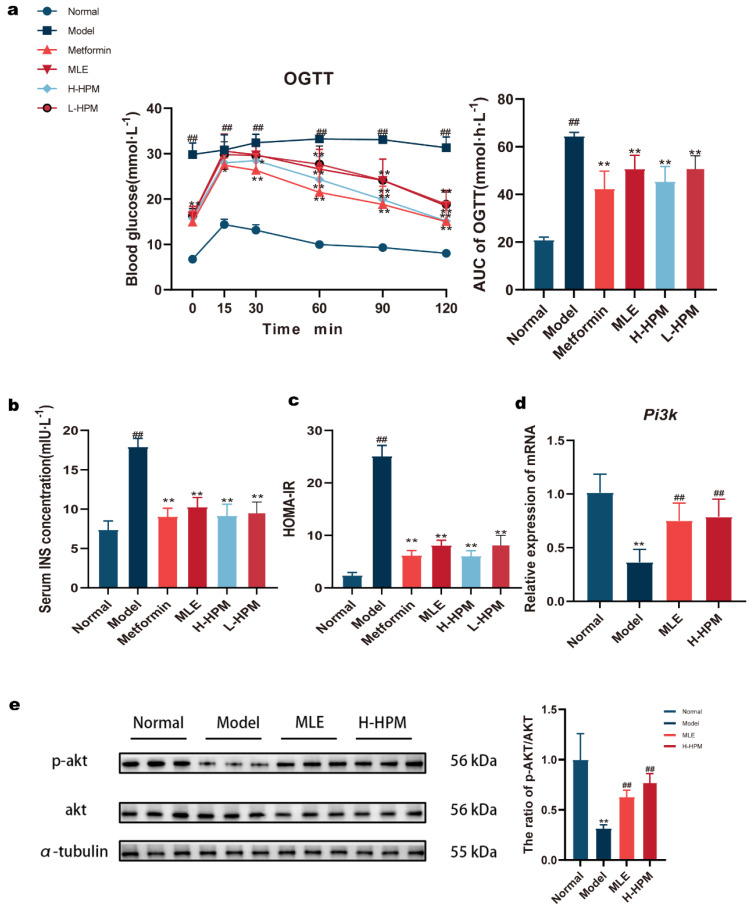
High-Protein Mulberry leaves actually enhance insulin sensitivity and modulate glucose metabolism. To study the role of High-Protein Mulberry leaves on glucose metabolism, OGTT was completed and the expression of glucose metabolism-related genes and proteins was measured. (**a**) OGTT and AUC. (**b**) Serum INS concentration. (**c**) HOMA-IR index. (**a**–**c**) * *p* < 0.05, ** *p* < 0.01 vs. Model group; ^##^ *p* < 0.01 vs. Normal group, (n = 7). (**d**) The relative mRNA levels of *Pi3k* in the liver. *β*-*actin* was used as an internal reference. (**e**) The ratio of p-Akt/Akt in the liver was measured by Western Blot and α-tubulin was used as a loading control. (**d**,**e**) ** *p* < 0.01 vs. Normal group; ^##^ *p* < 0.01 vs. Model group.

**Figure 4 ijms-25-08726-f004:**
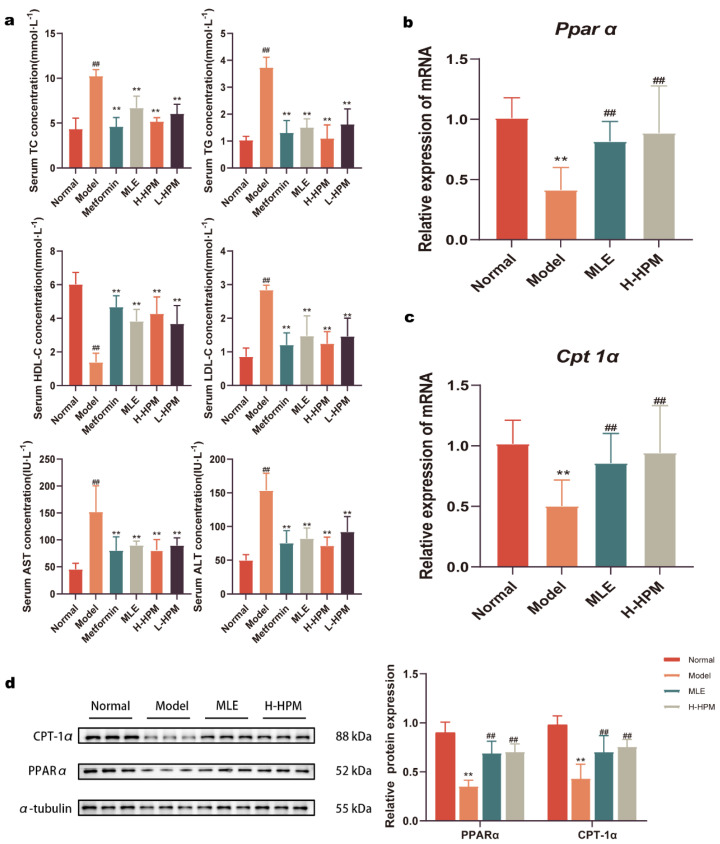
High-Protein Mulberry leaves effectively regulate lipid metabolism and improve liver function. Four items of blood lipid tests, ALT and AST were measured by biochemical kits. Moreover, the expression of genes and proteins related to lipid metabolism was quantified. (**a**) Biochemical analysis of TC, TG, HDL-C, LDL-C, ALT, and AST. ** *p* < 0.01 vs. Model group; ^##^ *p* < 0.01 vs. Normal group, (n = 7). (**b**,**c**) The relative mRNA levels of *Pparα* and *Cpt-1α* in the liver. *β-actin* was used as an internal reference. (**d**) The relative protein levels of CPT-1α and PPARα in the liver measured by Western Blot and α-tubulin was used as a loading control. ** *p* < 0.01 vs. Normal group; ^##^ *p* < 0.01 vs. Model group.

**Figure 5 ijms-25-08726-f005:**
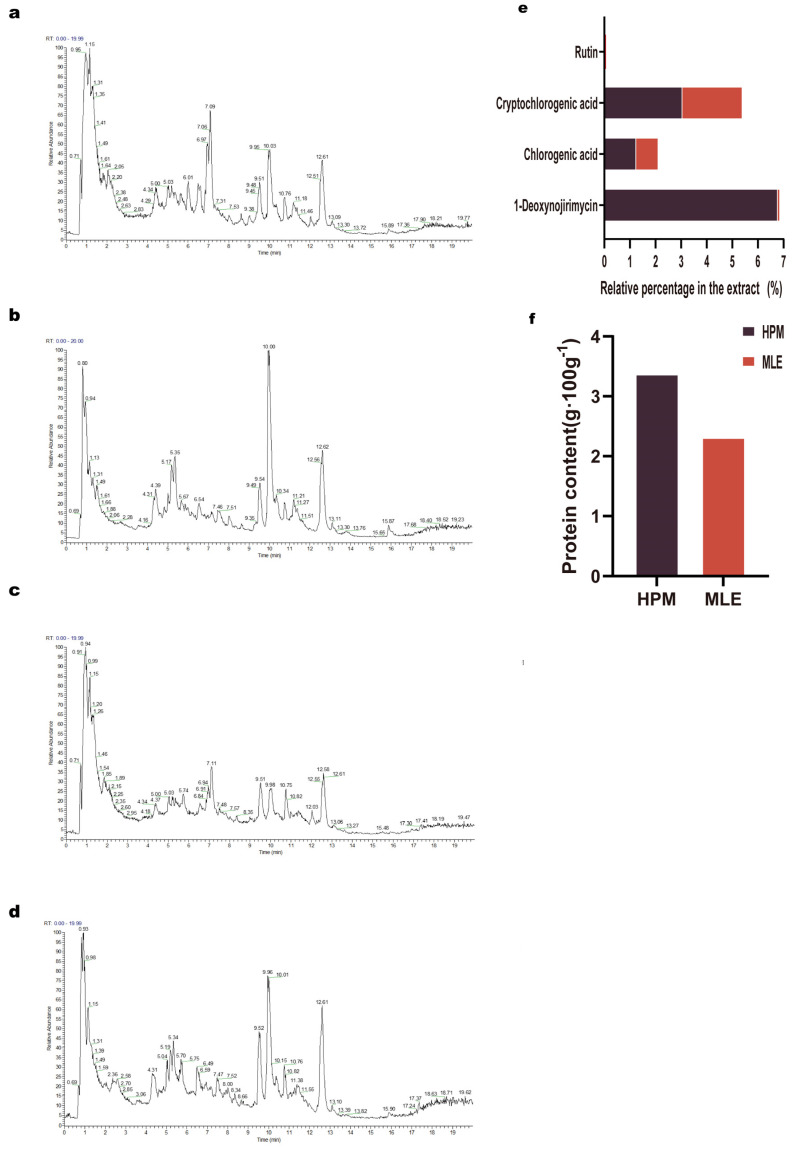
High-Protein Mulberry leaves are abundant in their composition and exhibit a significant relative content of the primary pharmacodynamic material basis. Differences in composition and content of High-Protein Mulberry leaves and Traditional Mulberry leaves were determined by UHPLC-MS and Kjeldahl. (**a**) Positive ion mass spectrometry of High-Protein Mulberry leaves. (**b**) Negative ion mass spectrometry of High-Protein Mulberry leaves. (**c**) Positive ion mode mass spectrometry of Traditional Mulberry leaves. (**d**) Negative ion mode mass spectrometry of Traditional Mulberry leaves. (**e**) Percentage of each major pharmacophoric component in the aqueous extract in UHPLC-MS. (**f**) protein content as determined by Kjeldahl method.

**Figure 6 ijms-25-08726-f006:**
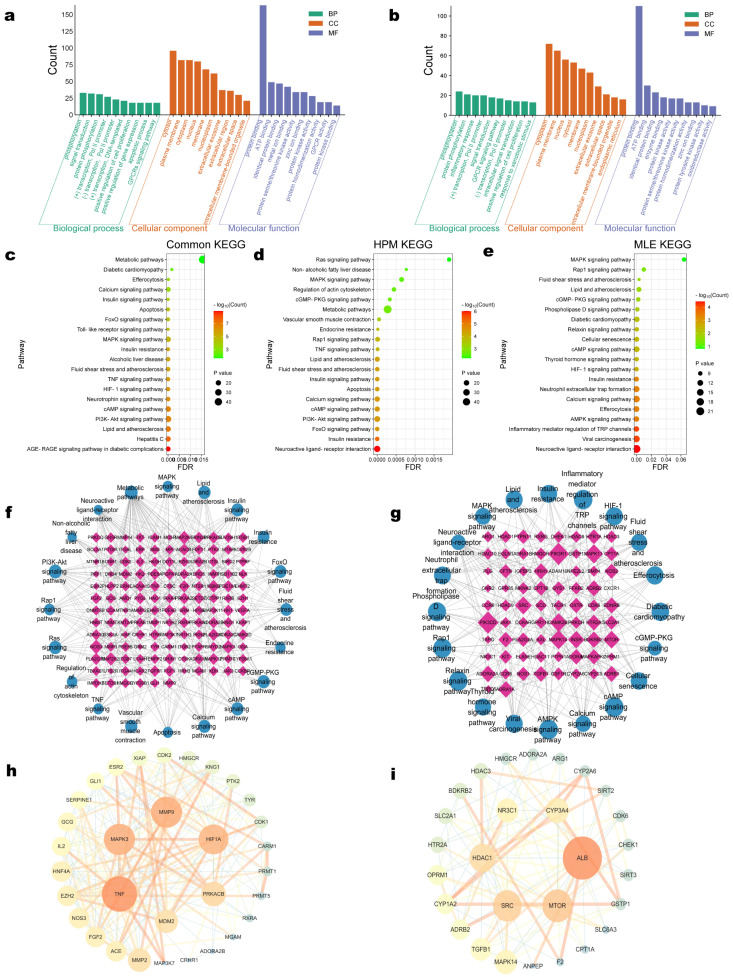
The differential components of High-Protein Mulberry leaves are associated with a larger total number of genes involved in T2DM and obesity-related pathways. (**a**) GO enrichment analysis of High-Protein Mulberry leaves. (**b**) GO enrichment analysis of Traditional Mulberry leaves. (**c**) KEGG bubble chart of common components. (**d**) KEGG bubble chart of High-Protein Mulberry leaves. (**e**) KEGG bubble chart of Traditional Mulberry leaves. (**f**) Target-pathway interaction network of High-protein Mulberry leaves. (**g**) Target-pathway interaction network of Traditional Mulberry leaves. (**h**) Core targets of High-Protein Mulberry leaves’ differential components. (**i**) Core targets of Traditional Mulberry leaves’ differential components.

**Table 1 ijms-25-08726-t001:** Primer sequences.

Primer	Forward	Reverse
*Pi3k*	CCACGACCATCTTCGGGTG	ACGGAGGCATTCTAAAGTCACTA
*Pparα*	CCAGCAACAACCCGCCTTTT	GGAGAGCCCGCATTACCTAC
*Cpt-1α*	ATCAAGAAGTGCCGGACGAG	GAACCTCTGCTCTGCCGTTG
*β-actin*	GCTACAGCTTCACCACCACA	AGGAAGGCTGGAAAAGAGCC

## Data Availability

The original contributions presented in the study are included in the article/Appendix A, further inquiries can be directed to the corresponding author.

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
