# Peer review of "High-Protein Mulberry Leaves Improve Glucose and Lipid Metabolism via Activation of the PI3K/Akt/PPARα/CPT-1 Pathway"

_ijms, 2024, doi:10.3390/ijms25168726_

Round 1

Reviewer 1 Report

Comments and Suggestions for Authors

This study investigated the effects of extracts of Mulberry leaves to treat type 2 diabetes. There was improvement in glucose tolerance by activating the mechanism that transmits the insulin signal. The study was well designed with the small number of mice in each group (7) being the only minor issue of concern. The clinical utility of these findings should be discussed. Mulberry trees are grown all over the world but the preparation of the extract to treat diabetes is not simple while metformin is cheap and easily available. Combining metformin and the mulberry leaf extract would be a good experiment to do in the future.

Minor points

The writing in figure 6 is very small so the figure is difficult to read.

Page 11 Line 280 . The statement “In these cases, the only way to prevent worsening of glucose tolerance is to increase insulin secretion, ie compensatory insulin secretion.” This statement is inaccurate. The best way to reverse glucose intolerance is to achieve weight loss. This has been shown with both surgical weight loss (Dixon JB et al. Adjustable gastric banding and conventional therapy for type 2 Diabetes. JAMA 299:316-323  2008) and by dietary means (Lean MEJ et al. Primary care-led weight management for remission of type 2 diabetes (DIRECT): an open label, cluster-randomised trial. The Lancet 391: 541-551. 2018).

It is better to say “ In these cases one way to prevent worsening……”

Page 12 Line 336.  “Enhancing insulin resistance” means making insulin resistance worse. I think the authors mean that insulin resistance was improved so The phrase should be “reducing insulin resistance..”  

Page 12 Line 350 What does “decoction” mean please explain what is meant.

Page 13 Line 380 It is stated that the mice were fasted for 12 hours to do the OGTT. A study has shown that the best time to fast mice to do these tests is 6 hours  (Andrikopoulos S et al Evaluating the glucose tolerance test in mice  AM J Physiology Endo and Metab 295  E1323-E1332 2008) so  the differences could have been more impressive.

Reviewer 2 Report

Comments and Suggestions for Authors

There has not been sufficient study in the West of plant based treatments for diabetes.  In China the role of traditional medicine affords the opportunity  to explore herbal medications.  Over the past decade, serious scientific efforts to analyze the biochemical changes underlying benefit of plant derived treatments have increased in China. This article deals with a new high protein mulberry variant.  The  demonstration of improvement in glucose insulin, lipid, and liver dynamics with administration of this extract in the db/db mouse model is through and convincing.  Of particular value is the utilization of both low protein mulberry leave extract and metformin as control conditions. the elucidation of the chemical components and comparison with low protein mulberry leaves is very nice.

One issue of concern, it is very understandable that metformin is used as a standard control, but if there is a major effect on PPAR pathways then pioglitazone should be used as a control agent.   Not asking to go back and repeat all the experiments now, but mentioned in the discussion with a view to the next set of experiments.

It is of interest to know the degree of use of mulberry leaves in China to treat diabetes.
